# Natural Rubber/Hexagonal Mesoporous Silica Nanocomposites as Efficient Adsorbents for the Selective Adsorption of (−)-Epigallocatechin Gallate and Caffeine from Green Tea

**DOI:** 10.3390/molecules28166019

**Published:** 2023-08-11

**Authors:** Kamolwan Jermjun, Rujeeluk Khumho, Mookarin Thongoiam, Satit Yousatit, Toshiyuki Yokoi, Chawalit Ngamcharussrivichai, Sakdinun Nuntang

**Affiliations:** 1Industrial Chemistry Innovation Program, Faculty of Science, Maejo University, Chiang Mai 50290, Thailand; kamol.nm@hotmail.com; 2Department of Chemical Technology, Faculty of Science, Chulalongkorn University, Bangkok 10330, Thailand; rujeeluk_fai_@hotmail.com (R.K.); mookarinth@gmail.com (M.T.); y.satit@hotmail.com (S.Y.); chawalit.ng@chula.ac.th (C.N.); 3Chemical Resources Laboratory, Tokyo Institute of Technology, Yokohama 226-8503, Japan; yokoi@cat.res.titech.ac.jp; 4Center of Excellence on Petrochemical and Materials Technology (PETROMAT), Chulalongkorn University, Bangkok 10330, Thailand; 5Center of Excellence in Catalysis for Bioenergy and Renewable Chemicals (CBRC), Faculty of Science, Chulalongkorn University, Bangkok 10330, Thailand

**Keywords:** natural rubber, hexagonal mesoporous silica, adsorption, (−)-epigallocatechin, caffeine

## Abstract

(–)-Epigallocatechin gallate (EGCG) is a bioactive component of green tea that provides many health benefits. However, excessive intake of green tea may cause adverse effects of caffeine (CAF) since green tea (30–50 mg) has half the CAF content of coffee (80–100 mg). In this work, for enhancing the health benefits of green tea, natural rubber/hexagonal mesoporous silica (NR/HMS) nanocomposites with tunable textural properties were synthesized using different amine template sizes and applied as selective adsorbents to separate EGCG and CAF from green tea. The resulting adsorbents exhibited a wormhole-like silica framework, high specific surface area (528–578 m^2^ g^−1^), large pore volume (0.76–1.45 cm^3^ g^−1^), and hydrophobicity. The NR/HMS materials adsorbed EGCG more than CAF; the selectivity coefficient of EGCG adsorption was 3.6 times that of CAF adsorption. The EGCG adsorption capacity of the NR/HMS series was correlated with their pore size and surface hydrophobicity. Adsorption behavior was well described by a pseudo-second-order kinetic model, indicating that adsorption involved H-bonding interactions between the silanol groups of the mesoporous silica surfaces and the hydroxyl groups of EGCG and the carbonyl group of CAF. As for desorption, EGCG was more easily removed than CAF from the NR/HMS surface using an aqueous solution of ethanol. Moreover, the NR/HMS materials could be reused for EGCG adsorption at least three times. The results suggest the potential use of NR/HMS nanocomposites as selective adsorbents for the enrichment of EGCG in green tea. In addition, it could be applied as an adsorbent in the filter to reduce the CAF content in green tea by up to 81.92%.

## 1. Introduction

(−)-Epigallocatechin gallate (EGCG) is a valuable bioactive component that is present at the highest concentration in green tea (*Camellia sinensis* L.). It is widely used in food, medicine, health products, and cosmetics because of its many health benefits, such as anticancer and anti-inflammatory effects [1,2], strong antimicrobial activity [3], antiviral effects [4], and prevention of HIV-1 infection [5]. EGCG is often purified from green tea by conventional methods, such as inorganic ion precipitation and solvent extraction. However, the EGCG extracted from green tea leaves using these technologies is contaminated with caffeine (CAF) [6,7]. Excessive CAF intake may cause adverse effects, including sleep deprivation, increased risk of cardiovascular disease, reduced fertility, and increased incidence of miscarriage [8,9]. According to regulations issued by the US Food and Drug Administration, the concentration of CAF as an ingredient in food and beverages should be limited to 200 parts per million (ppm) (0.02%) [10]. Therefore, an effective, industry-operational preparation method should be developed to obtain enriched EGCG green tea extract with as little CAF content as possible for use in pharmaceuticals, functional foods, and beverages.

The chemical structure of EGCG consists of C6–C3–C6 with two aromatic rings and several hydroxyl groups, whereas CAF is an N-containing fused heterocyclic system with two carbonyl groups (Figure 1). Conventionally, EGCG extract for food industries is obtained from green tea via inorganic ion precipitation or organic solvent extraction due to their high EGCG separation efficiency [11,12]. Chloroform, methylene chloride, methanol, and ethanol are commonly used in solvent extraction. However, these processes have many disadvantages, such as their use of large amounts of organic solvents or inorganic salts and the high toxicity of organic solvents. Moreover, the residual solvents in EGCG extract are highly unsafe because of their carcinogenic effects. Extraction methods using supercritical CO_2_ [13,14], simulated moving bed chromatography [7,15], and high-speed countercurrent chromatography [16,17] have been developed to obtain high-purity EGCG extract without harmful residues. However, these methods have high operation costs due to their expensive equipment and process complexity.

Mesoporous silica materials are widely used as adsorbents to purify bioactive compounds due to their high specific surface area, large pore volume, and narrow distribution of mesoscale pores. Pure-silica mesoporous materials display distinguishing features in this application, such as high purification efficiency, simple procedure, good stability, and easy regeneration [18,19,20,21,22]. Recently, EGCG was purified from green tea using mesoporous silica SBA-15 functionalized with aminosilane [23]. Although surface functionalization effectively enhances the adsorption selectivity of silica-based materials, it adds a process, cost, and waste discharge to adsorbent preparation. Mesoporous organic/silica hybrid materials are a class of nanocomposites that has been drawing considerable attention in the field of adsorption due to their high adsorption capacity and selectivity [24,25]. Natural rubber/hexagonal mesoporous silica (NR/HMS) is an interesting mesoporous silica-based nanocomposite; it consists of natural rubber (NR), which improves hydrophobicity, dispersed in the wormhole-like mesostructure of hexagonal mesoporous silica (HMS) [26,27]. NR/HMS materials have good structural and textural properties despite the simplicity of their reproducible preparation method, use of less harmful chemical reagents, and low cost. Moreover, the hydrophobicity of NR/HMS materials is crucial for controlled-release drug delivery [28].

In this work, we investigated the selective adsorption of (−)-epigallocatechin gallate and caffeine from green tea using a series of NR/HMS nanocomposites in comparison with pure-silica HMS materials. The effects of the mesopore size of these materials on EGCG and CAF adsorption capacity and selectivity were explored. An adsorption kinetic study, a simple but useful technique, was performed to gain a comprehensive understanding of EGCG and CAF adsorption on the NR/HMS nanocomposites. In addition, the performance of the NR/HMS materials in repeated adsorption–desorption of EGCG and CAF was investigated to verify the practical use of the nanocomposites in the selective separation of EGCG from green tea and to provide guidance for obtaining enriched EGCG green tea extract via adsorption. Moreover, a filter containing the NR/HMS adsorbent was developed for the reduction of CAF in green tea to produce healthy drinks.

## 2. Results and Discussion

### 2.1. Physicochemical Properties of Synthesized Adsorbents

The physicochemical properties of pure-silica HMS and the NR/HMS nanocomposites synthesized using different amine molecules are summarized in the Appendix A. The XRD patterns of pure-silica HMS and the NR/HMS nanocomposites synthesized using different amine molecules are presented in Appendix A. These materials exhibited a wormhole-like hexagonal mesostructure, as evidenced by the (100) diffraction peak appearing in the 2θ range of 1.0–3.0°. For the same type of amine template, the NR/HMS materials had a broader, less intense peak than the HMS series because the NR molecules in the synthesis mixture disturbed the self-assembly of surfactant micelles, thereby reducing silicate condensation [26,27]. In addition, the HMS-C_14_ and NR/HMS-C_14_ nanocomposites synthesized with the longest alkyl chain template had an enhanced degree of hexagonal packing of micelles, which resulted in a well-ordered hexagonal arrangement. This result agrees with a previous report [27].

FTIR analysis was used to evaluate the presence of NR in the HMS structure of the NR/HMS composites. As shown in Appendix A. The NR/HMS-C_12_ was used as a representative sample to compare its chemical structure with NR and pure-silica HMS-C_12_. The NR/HMS-C_12_ composite was observed between 1000 and 1300 cm^−1^, which represented the Si–O–Si stretching of the silica framework and exhibited a bord band at 3450 cm^−1^ related to free silanol groups (Si-OH) similar to HMS-C_12_. In addition, the bands that corresponded to characteristics of the NR structure at 2949, 2916, 2847, 1440, and 1365 cm^−1^ were observed. These results could suggest that the surface of HMS was partially covered by NR molecules.

Appendix A displays the typical N_2_ adsorption–desorption isotherms of pure-silica HMS and NR/HMS materials after extraction to remove the amine templates. These materials displayed type IV isotherms (according to IUPAC classification), which are characteristic of framework-confined mesoporous materials, and exhibited high S_BET_ values (Appendix A). The pore diameters of both synthesized materials were controlled by the template chain length (Appendix A inset), and V_t_ increased with the template size (Appendix A). By contrast, the NR/HMS series exhibited a lower N_2_ adsorbed volume and broader pore size distribution than HMS for the same amine template type (Appendix A), probably due to the presence of partially entrapped rubber chains inside the mesopores [26,27].

The hydrophobicity of these materials was represented by the monolayer adsorption volume (V_m_) at standard temperature and pressure, as determined via H_2_O adsorption–desorption measurement (Appendix A). The pure-silica HMS materials synthesized with larger amine molecules exhibited decreasing V_m_ values due to their reduced S_BET_. The NR/HMS materials exhibited lower V_m_ values than their HMS counterparts. This was attributed to not only the decreased S_BET_ values of the nanocomposites but also the hydrophobic environment created by the rubber phase exposed on the material surfaces [26,27].

The morphology of the HMS-C_14_ and NR/HMS-C_14_ nanocomposite is shown in representative FE-SEM images in Figure 2. HMS-C_14_ revealed very small spherical aggregates of dispersed silica particles as shown in Figure 2A. The presence of NR in mesoporous silica structure increased the agglomeration of mesoporous silica particles (Figure 2B). As a result, the mean particle size and interparticle porosity of NR/HMS-C_14_ were larger than that of HMS-C_14_. In addition, the interparticle porosity of NR/HMS was enhanced.

### 2.2. Adsorption of EGCG and CAF on Pure-Silica HMS and NR/HMS Materials

Figure 3 shows the EGCG and CAF adsorption capacity of both material series in the simulated solutions. Their EGCG adsorption capacity was higher than their CAF adsorption capacity, indicating that both pure-silica HMS and NR/HMS adsorbents favored the adsorption of EGCG rather than CAF (Figure 3). Although the NR/HMS adsorbents had a lower S_BET_ than the pure-silica HMS, the amounts of EGCG and CAF adsorbed on these nanocomposites were higher than those on the HMS materials. Ma et al. [23] reported that EGCG adsorption is driven by the combined action of hydrophobic interaction and hydrogen bonding, whereas CAF adsorption is driven by hydrophobic interaction only [23]. Therefore, the hydrophobicity of the NR/HMS nanocomposites had a positive effect and a large contribution to these materials’ adsorption of EGCG and CAF molecules. Zhang et al. [29] modified a silica surface with polyvinyl alcohol to improve its hydrophobicity. The resulting adsorbents exhibited a high adsorption capacity for tea polyphenols due to their enhanced hydrophobic interaction. The competitive adsorption of EGCG and CAF from the green tea extract solutions by both materials is shown in Figure 4. EGCG adsorption had higher selectivity than CAF adsorption. Notably, both the pure-silica HMS and NR/HMS series exhibited higher EGCG and CAF adsorption capacity compared with adsorption in single-compound systems (Figure 3). A similar observation was reported by Zhao et al. [30]. This result suggests a synergistic effect of the mixed solutes caused by the complexation of EGCG and CAF in the green tea solutions.

As shown in Figure 1, EGCG is a large bioactive molecule with three aromatic rings and several hydroxyl groups, whereas CAF is a relatively small heterocyclic compound. The pore diameter of solid sorbents is a crucial factor that determines their adsorption capacity and the diffusion of guest molecules. The EGCG and CAF adsorption capacity of the pure-silica HMS and NR/HMS materials were plotted against their pore diameters (Figure 5) to assess the effect of mesopore size on adsorption. EGCG and CAF adsorption capacity increased with the mesopore diameters of the pure-silica HMS and NR/HMS materials. The HMS-C_14_ and NR/HMS-C_14_ samples synthesized with the largest amine molecules exhibited the highest EGCG adsorption capacity (23.50 mg/g and 32.41 mg/g, respectively). One possible explanation is that the larger pore sizes of the adsorbents enhanced the adsorption due to the increase in pore volume with the amine template size (Appendix A).

### 2.3. Adsorption Behavior and Mechanism

#### 2.3.1. Adsorption Kinetics

The kinetic curves of EGCG and CAF adsorption on the NR/HMS-C_14_ adsorbent were investigated to elucidate the adsorption mechanism and rate (Figure 6). The adsorption mechanism depended on the physical and/or chemical characteristics of the adsorbent. EGCG and CAF adsorption capacity reached equilibrium after 250 min and 300 min, respectively. In 0–80 min, both EGCG and CAF adsorption capacity increased steeply, during which the adsorbate molecules diffused into the mesopore channels of the adsorbent. Afterward, both EGCG and CAF were adsorbed on the adsorbent via interparticle adsorption.

The experimental data of EGCG and CAF adsorption were fitted with the pseudo-first-order, pseudo-second-order, and intraparticle diffusion kinetic models, as shown in Figure 6. According to the calculated kinetic parameters (Table 1), the pseudo-first-order and intraparticle diffusion models exhibited lower R^2^ values than the pseudo-second-order model. The pseudo-second-order kinetic model was selected as the optimal model for describing EGCG and CAF adsorption on the NR/HMS-C_14_ adsorbent due to its high correlation coefficient and the affinity between its theoretical adsorption capacities and the experimental adsorption capacities. This model was developed from the assumption that an adsorbate binds strongly to an adsorbent surface, implying that the pseudo-second-order kinetic model should be applicable to systems involving electronic sharing or electron transfer between adsorbates and adsorbents. Therefore, the pseudo-second-order model was suitable for describing the adsorption kinetic behavior.

The equilibrium concentration of the total adsorbed EGCG (76.24 mg/g) was almost 29 times higher than that of the adsorbed CAF (2.65 mg/g) on the NR/HMS-C_14_ adsorbent. Adsorption selectivity was indicated by the selectivity factor (α), which was expressed as α = [q_e(EGCG)_/q_e(CAF)_] × [C_e(CAF)_/C_e(EGCG)_] [31]. Here, q_e(EGCG_ and q_e(CAF)_ (mg/g) are the equilibrium EGCG and CAF adsorption capacity, respectively, and C_e(EGCG)_ and C_e(CAF)_ (mg/mL) are the equilibrium concentrations of EGCG and CAF in the solutions, respectively. The adsorption selectivity coefficient of total EGCG versus CAF by the NR/HMS-C_14_ adsorbent was around 3.6, which was considerably higher than that of the other synthesized adsorbents (2–3). Therefore, the NR/HMS-C_14_ adsorbent preferentially adsorbed EGCG.

#### 2.3.2. Adsorption Isotherms

Adsorption isotherms characterize the distribution of adsorbate molecules on an adsorbent to identify the adsorption mechanism when adsorption equilibrium is reached at a constant temperature. The equilibrium adsorption capacity (q_e_) of EGCG and CAF on NR/HMS-C_14_ at different equilibrium concentrations (C_e_) was obtained at room temperature and is plotted in Figure 7. The Langmuir and Freundlich adsorption models were fitted with the experimental data to select the more suitable model. The Langmuir isotherm assumes that adsorption occurs at specific homogeneous sites within an adsorbent. The Freundlich isotherm is usable for a heterogeneous adsorbent surface with a non-uniform distribution of heat of adsorption over the surface. From the experimental and predicted adsorption data (Figure 7), the Langmuir model fit EGCG and CAF adsorption on the NR/HMS-C_14_ nanocomposite better as it had higher R^2^ values (Table 2). Thus, EGCG and CAF were adsorbed on the homogeneous sites on the adsorbent surface. These results were consistent with those of Ma et al. [23], who reported that the adsorption isotherm of EGCG adsorbed on SBA-15-NH_2_ fitted the Langmuir model.

However, these isotherm studies suggested that the EGCG and CAF adsorption onto the surface of NR/HMS-C_14_ proceeded via a combination of different interaction forces. Although the adsorption of EGCG and CAF mainly provided the hydrogen bonding interaction from silanol groups on the mesoporous silica surface, it was also influenced by the hydrophobicity of the added natural rubber.

#### 2.3.3. Adsorption Mechanism of EGCG and CAF on Mesoporous Silica Surface

The surface chemistry of silica before and after adsorption was elucidated to gain useful information for assessing the adsorption mechanism. The major components of the HMS and NR/HMS materials were their silica content, and their surfaces had large quantities of silanol groups, as reported in previous studies [26,27]. Therefore, pure-silica HMS-C_14_ was used as a representative sample to study the adsorption phenomena while avoiding the effect of other organic matters on the adsorption of EGCG and CAF on the silica surface.

In this study, HMS-C_14_ was used as a representative pure-silica adsorbent. The XPS samples were prepared by adding 0.1 g of the adsorbent to 10 mL of an EGCG or CAF solution (150 ppm) and stirring for 12 h at room temperature. As shown in Figure 8, the wide-scan XPS results of pristine HMS and HMS-adsorbed EGCG and CAF (HMS-C_14_-EGCG and HMS-C_14_-CAF, respectively) exhibited the main signals of the Si, C, and O species (Figure 8A). A low-intensity N signal was found in HMS-C_14_-CAF due to the N atoms of the CAF molecule.

From the C1s signal deconvolution (Figure 8B), the fresh HMS exhibited signals at 284.4 eV and 285.5 eV, which were assigned to C–C/C–H and C–O bonds, respectively. The presence of organic species was related to the residual nonhydrolyzed ethoxy groups after the sol–gel synthesis [27]. After the adsorption of EGCG and CAF on HMS-C_14_, the C1s signal increased. HMS-C_14_-EGCE had five signals at 284 eV, 284.6 eV, 285.7 eV, 287.9 eV, and 288.8 eV; these were attributed to the C=C, C–C/C–H, C–O, C=O, and O–C=O bonds, respectively, which constituted the chemical structure of the EGCG molecule [32,33]. HMS-CAF exhibited five signals at 284.6 eV, 285.1 eV, 285.9 eV, 287 eV, and 288.8 eV, which corresponded to the C–C/C–H, C=N, C–O, C=O, and N–C=O bonds, respectively, of the CAF molecule [34,35].

The O1s XPS spectra of pure-silica HMS revealed deconvoluted signals at 531 eV, 532.9 eV, and 533.9 eV (Figure 8C), corresponding to the Si–O–C, Si–O–Si, and Si–O–H species, respectively [27]. After adsorption, additional signals were observed at 530 eV and 531.9 eV, which were attributed to the C–O–H and C=O bonds, respectively, of the adsorbed EGCG and CAF molecules. The overall results suggest that there was no change in the chemical structure of the adsorbed EGCG and CAF molecules. Therefore, EGCG and CAF, with carbonyl and/or carboxyl groups, were adsorbed on the silica surface via possible hydrogen bonding with silanol groups. This phenomenon was similar to the case of EGCG and CAF adsorbed on the SBA-15 material reported by Ma et al. [23].

The mechanism of EGCG and CAF adsorption on HMS or NR/HMS adsorbents is postulated in Figure 1. It accounts for only the interaction of the adsorbate molecules with the silanol groups of the silica surface. Furthermore, the EGCG molecule possessed a higher polarity than the CAF [14]. These results indicated a higher adsorption capacity for EGCG than CAF on both materials since hydrogen bonding occurred between EGCG and mesoporous silica HMS or NR/HMS due to the large number of hydroxyl groups in the EGCG structure.

### 2.4. Desorption Capacity and Desorption Ratio

The EGCG and CAF adsorbed on the NR/HMS-C_14_ adsorbent were desorbed by different eluents in ethanol/deionized water solutions with various (*v/v*) ratios. The desorption efficiency of the different eluents is shown in Figure 9. According to the experimental data, the eluent where deionized water had a higher polarity than ethanol exhibited a lower desorption capacity and desorption ratio in the removal of EGCG and CAF from the NR/HMS-C_14_ surface. In addition, an increase in the amount of deionized water in the tested eluents reduced the desorption efficiency of both adsorbates. However, the eluent with the alcohol/deionized water ratio of 80:20 (*v/v*) had the highest desorption capacity. These results agreed with those of Fan et al. [36]. Protic solvents with high polarity, such as water, can act as both a donor and acceptor of hydrogen bonds and easily associate with each other, although they also possess high hydrogen bond potentials, which may weaken their interactions with adsorbent/adsorbate complexes and reduce EGCG and CAF desorption efficiency [36].

The NR/HMS-C_14_ adsorbent exhibited a higher desorption efficiency of EGCG than CAF compared with the eluent with the same alcohol/water ratio. This result was related to the adsorption capacity and behavior of EGCG and CAF on the NR/HMS materials (Figure 3 and Figure 4). The preferential adsorption of the NR/HMS materials for EGCG resulted in a larger amount of EGCG desorbed from the adsorbent surface.

### 2.5. Reusability

Studies on EGCG and CAF desorption from the NR/HMS-C_14_ surface exhibited the effectiveness of the eluent with the ethanol/deionized water solution ratio of 80:20 (*v/v*). The reusability of the NR/HMS-C_14_ adsorbent was preliminarily evaluated under the same adsorption conditions, as shown in Figure 10. The spent adsorbent was recovered from the EGCG or CAF solution by filtration, thoroughly washed with ethanol, and dried at room temperature. The amounts of EGCG and CAF adsorbed in the second cycle were slightly smaller. The adsorbent could be repeatedly used for adsorption at least three times. Adsorption capacity loss (~70 wt.%) was observed in the fourth repetition. This might have been due to the strong adsorption of organic substances on the NR/HMS surface.

### 2.6. Application to Reduce the CAF in Green Tea

In general, green tea has less CAF than EGCG. However, CAF has a negative effect on the health of some consumers. Therefore, a simple experiment was conducted to verify the effectiveness of NR/HMS adsorbent for reducing CAF in green tea to produce healthy drinks. We developed filters containing different amounts of NR/HMS nanocomposites, such as 0.5, 1.0, and 1.5 g of filters A, B, and C, respectively (Figure 11). An amount of 30 mL of green tea containing approximately 104 mg/L of CAF was poured through a filter. Then, the filtrate was analyzed using HPLC to determine the amount of CAF in green tea after filtration. The study found that the percent removal of CAF in green tea using the three filters was ranked in the following descending order: filter C > filter B > filter A, which corresponded to the amount of the NR/HMS contained in the filter (Figure 11). Moreover, filter C exhibited a maximum average caffeine removal of 81.92% per use.

## 3. Experimental

### 3.1. Materials and Chemical Reagents

Tetraethyl orthosilicate (TEOS; AR grade 99%) and primary amines, namely, decylamine (C_10_; AR grade 99%), dodecylamine (C_12_; AR grade > 99%), and tetradecylamine (C_14_; AR grade 95%), were purchased from Sigma-Aldrich (Darmstadt, Germany). Tetrahydrofuran (THF; AR grade 99.5%) was obtained from QREC Chemicals Co. Ltd. (Auckland, New Zealand). Sulfuric acid (H_2_SO_4_; AR grade 98%) and absolute ethanol (AR grade 99.5%) were purchased from Merck Millipore Ltd (Darmstadt, Germany). Technically specified NR (standard Thai rubber, grade 5L) was supplied by the Thai Hua Chumporn Natural Rubber Co. Ltd. (Bangkok, Thailand). Commercially available dried green tea leaves were obtained from Royal Project Tea (Chiang Mai, Thailand). The EGCG (AR grade 98%) standard was purchased from Word-Way Biotech Inc. (Hunan, China). CAF (AR grade 98%) was purchased from Sigma-Aldrich (Darmstadt, Germany). All materials and reagents were used without further purification.

### 3.2. Preparation of Adsorbents

#### 3.2.1. Synthesis of Pure-Silica HMS Materials

A series of siliceous HMS adsorbents with tunable textural properties were synthesized by modifying the type of primary amine template as previously reported [27]. The molar composition of the synthesis mixture was 0.10 TEOS:0.04 amine:5.89 H_2_O:0.37 THF. Typically, the primary amine was dissolved in THF, and deionized water was slowly added via constant stirring. After 1 h, TEOS was added dropwise, and the resulting mixture was vigorously stirred for 0.5 h at 40 °C and then aged at ambient temperature for 18 h. Subsequently, the white solid was recovered by filtration, thoroughly washed with deionized water, and dried at 100 °C overnight to obtain as-synthesized HMS. Finally, the amine template was removed by refluxing 3 g of the as-synthesized HMS with 150 mL of a 0.05 M H_2_SO_4_/ethanol solution for 4 h. The generated HMS materials were labeled HMS-C_n_, where n is the number of carbon atoms of the primary amine.

#### 3.2.2. Synthesis of NR/HMS Nanocomposites

NR/HMS nanocomposites were synthesized using different primary amine templates via an in situ sol–gel process [27]. In a typical synthesis, an NR sheet (0.5 g) was saturated with TEOS (10 g) at room temperature for 16 h. The resulting swollen NR was weighed to determine the TEOS content (~1.8 g) and then dissolved in THF through vigorous stirring overnight. A primary amine and an additional amount of TEOS (~8.7 g) were slowly added to this colloidal solution via constant stirring at 40 °C for 30 min. Subsequently, deionized water was added to the resulting mixture by stirring. The molar composition of the synthesized mixture was 0.10 TEOS:0.04 amine:5.89 H_2_O:0.37 THF:0.01 NR. After 1 h of stirring, the white gel was kept at 40 °C for 3 days and then precipitated in 100 mL of ethanol. The recovery of the solid product and the template removal procedure were the same as those described in Section 3.2.1. The template-free nanocomposites were named NR/HMS-C_n_, where n is the number of carbon atoms of the primary amine.

### 3.3. Characterization of Pure-Silica HMS and NR/HMS Materials

X-ray diffraction (XRD) patterns were obtained using a Rigaku SmartLab diffractometer with Cu Kα radiation (λ = 0.154 nm). The X-ray source was operated at 40 kV and 40 mA, and counts were accumulated every 0.02° (2θ) at a scan speed of 1° (2θ)/min.

The presence of functional groups in these materials was characterized using a Perkin Elmer Spectrum One Fourier-transform infrared spectrometer (FTIR). The sample wafer was prepared following the KBr method. The FTIR spectra were collected in transmittance mode between 400 and 4000 cm^−1^. Nitrogen adsorption–desorption isotherms were measured at −196 °C using a Micrometrics ASAP 2020 instrument. A template-free sample (~50 mg) was heated in a vacuum at 150 °C for 2 h for outgassing. The specific surface area (S_BET_) was calculated according to Brunauer–Emmett–Teller (BET) theory using adsorption data at relative pressures (P/P_0_) of 0.05–0.3. The total pore volume (V_t_) was evaluated at a relative pressure of about 0.99. The pore diameter was calculated from the desorption branches using the Barrett–Joyner–Halenda (BJH) method.

The material morphologies were studied by field emission scanning electron microscopy (FE-SEM). The FE-SEM images were recorded on a TESCAN CLARA scanning electron microscope operating at 20 kV. The samples on copper grids were observed with platinum coating.

The chemical composition of the adsorbent surface was investigated by X-ray photoelectron spectroscopy (XPS) using an ESCA 1700R system with Al Kα1 radiation (1486.8 eV). The C 1s signal at 284.6 eV, corresponding to adventitious carbon, was used to calibrate the binding energy scale. The XPS spectra were deconvoluted using the software OriginPro 8.5 prior to the quantification of the different bonding species.

Water adsorption–desorption was applied to investigate the hydrophobicity of pure-silica HMS and NR/HMS nanocomposites. Measurement was performed at 25 °C using a BEL Japan BELSORP-max. Each sample was pretreated in a similar manner to N_2_ physisorption. The monolayer adsorbed volume of water (V_m_) was obtained from adsorption data at a P/P_o_ of below 0.3.

### 3.4. Preparation of EGCG, CAF, and Green Tea Extract Solutions

EGCG and CAF solutions with different concentrations were prepared by dissolving a specific amount of EGCG or CAF in deionized water at 60 °C for 1 h. The green tea extract was prepared by refluxing dried green tea leaves (25 g) with 200 mL of ethanol/deionized water (60:40, *v/v*) solution at 90 °C for 1 h. The extraction step was repeated three times. The resulting mixture was filtered, and the solvent was removed by using a rotary evaporator at 50 °C at a reduced pressure. The obtained solid was dried in a vacuum oven at 50 °C until the mass was constant, ground to powder, and kept in a refrigerator at 4 °C. Finally, the extract powder was dissolved in deionized water to prepare green tea solutions with different concentrations.

### 3.5. Determination of EGCG and CAF Concentrations

High-performance liquid chromatography (HPLC) was used to determine the concentrations of EGCG and CAF in the solutions. Analysis was performed using an Agilent 1100 series HPLC system (Agilent Technologies, Waldbronn, Germany) equipped with a quaternary pump, an online degasser, a thermostatically controlled column compartment, and a diode array detector coupled with an analytical workstation. Chromatographic separation was conducted using an Agilent Eclipse XDB-C18 column (4.6 mm × 250 mm, 5 m) at 28 °C and a detection wavelength of 280 nm using 14–25% acetonitrile/water (pH 4 adjusted by H_3_PO_4_) as an eluent at a flow rate of 2 mL/min. The aqueous solutions of the samples and standards were filtered through 0.45 μm filters before use. The retention times of EGCG and CAF were 1.92 min and 2.78 min, respectively. Six concentration points were used to plot a calibration curve. The regression equations for the EGCG and CAF solutions were y = 2.1618x + 7.9094 (R^2^ = 0.9998) and y = 2.5126x + 5.7012 (R^2^ = 0.9995), respectively, where x is the concentration (mg/L) and y is the peak area.

### 3.6. Adsorption Procedure

The performance of pure-silica HMS and NR/HMS materials in EGCG and CAF adsorption was evaluated through static adsorption at room temperature. Typically, 10 mL of each sample solution (EGCG, CAF, and green tea extract) was mixed with 0.1 g of a dried absorbent in a 50 mL Erlenmeyer flask. The flask was sealed tightly and shaken on an incubation orbital shaker at 120 rpm for 12 h to ensure adsorption equilibrium. Subsequently, the solution was filtered to remove the adsorbent and subjected to HPLC analysis. The equation to calculate the adsorption capacity (q_e_) is described in Appendix A.

### 3.7. Adsorption Kinetic Study

The dried absorbent (1.0 g) was added to the EGCG or CAF solution (100 mL) in the 250 mL Erlenmeyer flask. The initial concentrations of the EGCG and CAF solutions were fixed at 100 mg/L and 50 mg/L, respectively. The other processes were the same as those described in Section 3.6. The EGCG and CAF solutions were withdrawn at different time intervals for HPLC analysis until equilibrium was reached. The equations of the kinetic models are described in Appendix A.

### 3.8. Adsorption Isotherm Study

The adsorption isotherm data were measured via the procedure described in Section 3.6. The initial concentrations of EGCG and CAF were varied in the range of 20–100 mg/L.

### 3.9. Desorption of Adsorbed EGCG and CAF and Reusability of Adsorbents

A static desorption test of the spent adsorbents was performed after the equilibrium of EGCG and CAF adsorption was reached. Ethanol/deionized water solutions with various (*v/v*) ratios were used as solvents, and the liquid:solid ratio was maintained at 50 mL: 1 g. The flasks were continually shaken at 120 rpm for 12 h at room temperature. The EGCG and CAF contents of the desorbed solutions were analyzed by HPLC. Each process was repeated three times. The equations to calculate the desorption capacity (q_d_) and desorption ratio (D) are described in Appendix A.

After the course of desorption, each adsorbent was recovered by filtration, thoroughly washed with ethanol to remove the organic residue on the surface and dried at room temperature. The regenerated adsorbents were reused in new batches containing fresh EGCG and CAF solutions under typical conditions. The reusability of the adsorbents was investigated until a significant decline in adsorption capacity was observed.

## 4. Conclusions

The proposed NR/HMS nanocomposites, which had diverse pore sizes and hydrophobicity, exhibited a higher EGCG and CAF adsorption efficiency than the pure-silica HMS series. The pore size and hydrophobicity of the mesoporous silica materials played critical roles in the highly efficient and selective adsorption of EGCG by the nanocomposite. In addition, the major driving force for the adsorption of EGCG and CAF on the mesoporous silica surface was hydrogen bonding. NR/HMS-C_14_, with its large pore size and hydrophobicity, showed the highest EGCG adsorption capacity (32.41 mg/g) among all the synthesized adsorbents. The kinetic data fitted well using a pseudo-second-order kinetic model, and the adsorption selectivity coefficient of EGCG was 3.6. The adsorption results of EGCG and CAF fitted the Langmuir model better than the Freundlich model. Large amounts of EGCG and CAF could be removed from the NR/HMS surface using a solution of ethanol and deionized water (80:20, *v/v*) as the solvent, and NR/HMS exhibited a higher desorption capacity for EGCG than CAF. Moreover, the NR/HMS materials could be reused three times for EGCG adsorption. These findings demonstrate that the NR/HMS nanocomposite can be an excellent EGCG adsorbent in aqueous solutions. It is a promising tool for the separation and enrichment of EGCG from complex natural products. It can also be applied as an adsorbent in a simple filter to reduce the CAF in green tea.

## Data Availability

The data presented in this study are available in the article and the Appendix A.

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
