# Peer review of "Natural Rubber/Hexagonal Mesoporous Silica Nanocomposites as Efficient Adsorbents for the Selective Adsorption of (−)-Epigallocatechin Gallate and Caffeine from Green Tea"

_molecules, 2023, doi:10.3390/molecules28166019_

Round 1

Reviewer 1 Report

In this MS, the authors described synthesis of templated silica (hexagonal mesoporous silica, HMS) alone and composites of silica with natural rubber (NB) characterized using several methods (SEM, XPS, XRD, nitrogen and water adsorption and tested on the adsorption of epigallocatechin gallate (EGCG) and caffeine (CAF) from green tea extract solutions. Obtained results are of interest from a practical point of view. However, the MS needs, at least, in double major revision.

(1)     Abstract: “...the hydroxyl groups of EGCG and CAF.” CAF does not have the hydroxyl groups.

(2)     Some information on routine calculation methods (e.g., adsorption, kinetics) described in Experimental could be placed in Supplementary information (SI) file.

(3)     The NR structure should be described in the SI file. At least, FTIR spectra could be recorded for NR, HMS, and NR/HMS and appropriately analyzed in the SI file.

(4)     Description of the water adsorptiondesorption (lines 159-163) is unclear (“The monolayer adsorbed volume of water (Vm) was obtained from adsorption data at a P/Po of below 0.2”???). The monolayer adsorption should be determined using, e.g., the Langmuir eq. It will be better to show the water ads-des isotherms in the SI file.

(5)     Chap. 3.1. Physicochemical properties of synthesized adsorbents is too short and low-informative. The analysis of the characteristics is incomplete. Effects of NR on the composite characteristics are poorly analyzed.

(6)     The textural characteristics (Table S1, PSD in Fig. S2) do not correspond to the nitrogen adsorption isotherms (Fig. 2) showed with large shits along the Y-axis. A similar presentation of the results is absolutely wrong.

(7)     The BJH method cannot be used to compute the PSD of the studied materials because it can give PSD only at dp > 2 nm. There are some additional errors of the BJH method that appear upon the analysis of similar materials. As a whole, there is no any sense to show the wrong PSD in the paper. The NLDFT of QSDFT methods are strongly recommended.

(8)     “...(BET) theory using adsorption data at relative pressures (P/P0) of 0.02–0.2.” This pressure range is incorrect for the BET analysis. There are several parameters which can be used for appropriate selection of this interval (see Pure Appl. Chem. 2015, 87(9–10), 1051), maybe 0.6-0.22 could be better.

(9)     The XRD patterns are not analyzed; however, some interesting information on the texture and morphology of the studied materials could be obtained.

(10)  Figures 3, 9, and 10 should be re-drawn similarly to Figure 4.

(11)  In many Figures, the values on the axes are much larger than the interval of experimental values. This is rather inappropriate method of data visualization.

(12)  Scheme 1 gives incomplete information on the adsorption complexes, e.g., CAF can form the hydrogen bonds with participation of the N atoms. The ads. complexes are rather multidentate with participation of several silanols per adsorbed molecule, especially EGCG.

It should be checked.

Author Response

Thank you for sending us the reviewers’ comments. We revised the manuscript according to the comments already. The changes made according to the comments were highlighted in blue.  

Reviewer 2 Report

1.- The IUPAC standard methodology for BET surface area estimation suggest to use relative pressures from 0.05 to 0.30. However, the authors used relative pressures from 0.02 to 0.2. Could the authors explain why they used a different range of relative pressures? 

2.- line 162: "...from adsorption data at a P/Po of below 0.2." Correct wording

3.- The quantities in Table S1 should be reported with errors.

4.- Axis of insets in Fig. S2 must have labels and titles.

5.- The pore size distribution of sample HMS-C12 contains too few points to draw conclusions from it. It is desirable to repeat the experiment with more points around the capillary condensation.

6.- line 307 "The larger pore sizes of the adsorbents not only facilitated the rapid diffusion of EGCG and CAF into the pore channels but also enhanced adsorption due to the increase in pore volume with the amine template size (Table S1)." There is insufficient evidence to make the claims that are made. At this point in the manuscript it has not been demonstrated that EGCG and CAF molecules adsorb on mesoporosity, for example. It could be the case that increasing the dimensions of the template molecules leads to a decrease in particle size and thus an increase in external surface area.

7.- From line 364 to the end of the paragraph: Please, check the mechanistic-statistical derivation of Langmuir-type models in this book (https://www.routledge.com/Adsorption-and-Diffusion-in-Nanoporous-Materials/Roque-Malherbe/p/book/9780367572167). The Langmuir model can describe adsorption in any homogeneous adsorption domain. The domain does not necessarily have to be a monolayer. In the case of the materials under study, there will most likely be some compensation between the obvious heterogeneities of the material and the lateral interactions between molecules as the pores fill. 

8.- section 3.3.3: If adsorption take place ONLY through the interaction of both adsorbate molecules with the silanol groups of the silica surface, how can the authors explain that the smaller molecule adsorbs less? 

The quality of the English used is good. Some minor details were pointed out in the previous section.

Author Response

Thank you for sending us the reviewers’ comments. We revised the manuscript according to the comments already. The changes made according to the comments were highlighted in green.  

Round 2

Reviewer 1 Report

Revised MS could be recommended for publication

English should be polished by a native speaker.